# Risk Factors of Glecaprevir/Pibrentasvir-Induced Liver Injury and Efficacy of Ursodeoxycholic Acid

**DOI:** 10.3390/v15020489

**Published:** 2023-02-09

**Authors:** Hideyuki Tamai, Jumpei Okamura

**Affiliations:** Department of Hepatology, Wakayama Rosai Hospital, 93-1 Kinomoto, Wakayama 640-8505, Japan

**Keywords:** hepatitis C virus, glecaprevir, pibrentasvir, drug-induced liver injury, ursodeoxycholic acid

## Abstract

Although glecaprevir/pibrentasvir (GP) therapy is recommended as a first-line treatment for hepatitis C virus (HCV) infection, serious drug-induced liver injury occasionally develops. The present study aimed to elucidate real-world risk factors for GP-induced liver injury and to evaluate the efficacy of add-on ursodeoxycholic acid (UDCA) for liver injury. We analyzed 236 HCV patients who received GP therapy. GP-induced liver injury was defined as any elevation to grade ≥ 1 in total bilirubin (TB), aspartate aminotransferase (AST), alanine aminotransferase (ALT), alkaline phosphatase (ALP), or γ-glutamyl transferase (γ-GT) during treatment without other cause. The frequency of GP-induced liver injury was 61.9% (146/236). Serious elevation to grade ≥ 3 in TB, AST, ALT, ALP, and γ-GT was identified in 3.8% (9/236), 0%, 0%, 0%, and 0.4% (1/209), respectively. Therapy discontinuation and dose reduction were seen in one patient each. Multivariate analysis revealed age and TB as independent risk factors for GP-induced liver injury. In patients with grade ≥ 2 hyperbilirubinemia, TB after onset significantly decreased in the add-on UDCA group but not in the no UDCA group. Careful attention to GP-induced liver injury is warranted for elderly patients with cirrhosis. Add-on UDCA could suppress the aggravation of GP-induced liver injury.

## 1. Introduction

In November 2017, the clinical use of pan-genotypic, dual direct-acting antiviral agent (DAA) therapy using the non-structural protein (NS)3/4A protease inhibitor glecaprevir and the NS5A inhibitor pibrentasvir was approved in Japan for hepatitis C virus (HCV)-infected patients without cirrhosis or with compensated cirrhosis [1,2]. Glecaprevir/pibrentasvir (GP) therapy appears to offer three advantages over other DAA regimens: (1) shorter treatment duration; (2) stronger genetic barrier to viral resistance; and (3) no restriction among patients with renal impairment or on dialysis, because these drugs are all metabolized hepatically. GP therapy is now recommended as one of the first-line, pan-genotypic DAA therapies for HCV patients without cirrhosis or with compensated cirrhosis, including non-responders to previous DAA therapy [3,4,5]. However, NS3 protease inhibitor-containing regimens are contraindicated in patients with decompensated cirrhosis and in patients with compensated cirrhosis with previous episodes of decompensation because the blood concentrations of these drugs are markedly higher in patients with decompensated cirrhosis [3,4,5]. In fact, although the GP regimen shows a safe profile of adverse effects even in real-world settings [6], serious liver injury develops in some patients, leading to therapy discontinuation or liver failure [7,8,9,10]. However, adequate methods for coping with GP-induced liver injury are still unclear, and no consensus has been reached on criteria for drug discontinuation, interruption, or dose reduction once the serious liver injury has developed. Serious GP-induced liver injury thus remains troubling for attending physicians. Risk factors for drug-induced liver injury represent crucial information in the selection of adequate DAA regimens, and optimal methods to deal with GP-induced liver injury must be established in the interest of the safe completion of therapy.

Ursodeoxycholic acid (UDCA) has long been used empirically as a drug for cholestatic drug-induced liver injury to shorten the time to resolution [11,12]. For daclatasvir/asunaprevir as an NS3 protease inhibitor-containing regimen, Taki et al. indicated that the lower frequency of serious drug-induced liver injury compared to previous reports may have been attributable to the use of add-on UDCA [13]. However, evidence for the efficacy of UDCA for protease inhibitor-induced liver injury remains insufficient.

The aim of the present study was to elucidate risk factors for GP-induced liver injury in a real-world setting and to evaluate the efficacy of add-on UDCA for such liver injury.

## 2. Materials and Methods

### 2.1. Patients

This was a retrospective cohort study of GP therapy for HCV in a real-world setting. Between January 2019 and May 2022, a total of 241 consecutive HCV patients without cirrhosis or with compensated cirrhosis received GP therapy in our hospital. No patient with acute hepatitis C was included in this study. Of those 241 patients, 4 patients who were lost to follow-up and 1 patient who died due to bacterial pneumonitis unrelated to drugs were excluded from the present investigation. Finally, 236 patients were enrolled in the present study for analysis.

### 2.2. Treatment and Follow-Up

A standard dose of GP therapy (glecaprevir at 300 mg/day; pibrentasvir at 120 mg/day) was orally administered for 8 weeks to patients without cirrhosis and for 12 weeks to patients with compensated cirrhosis. Baseline liver function was assessed using albumin-bilirubin (ALBI) grade [14]. HCV-RNA was examined using TaqMan PCR assay every month. Biochemical tests, including blood counts, were performed every two weeks during treatment, and safety and tolerability were assessed by attending physicians who monitored adverse events. The approach to dealing with adverse effects was decided by each attending physician. Sustained virological response (SVR) was defined as viral negativity 12 weeks after the end of therapy.

### 2.3. Assessment of GP-Induced Liver Injury

Because of the lack of a simple objective test for the diagnosis of drug-induced liver injury [15], GP-induced liver injury was defined as any elevation in total bilirubin (TB), aspartate aminotransferase (AST), alanine aminotransferase (ALT), alkaline phosphatase (ALP), or γ-glutamyl transferase (γ-GT) during treatment without any other evident cause. The grade of GP-induced liver injury was assessed according to Common Terminology Criteria for Adverse Events version 5.0. TB elevation was defined as follows: Grade 1 was >upper limit of normal (ULN)-1.5 × ULN if the baseline was normal; >1.0–1.5 × baseline if the baseline was abnormal. Grade 2 was >1.5–3.0 × ULN if the baseline was normal; >1.5–3.0 × baseline if the baseline was abnormal. Grade 3 was >3.0–10.0 × ULN if the baseline was normal; >3.0–10.0 × baseline if the baseline was abnormal. AST and ALT elevations were defined as follows: Grade 1 was >ULN-3.0 × ULN if the baseline was normal; 1.5–3.0 × baseline if the baseline was abnormal. Grade 2 was >3.0–5.0 × ULN if the baseline was normal; >3.0–5.0 × baseline if the baseline was abnormal. Grade 3 was >5.0–20.0 × ULN if the baseline was normal; >5.0–20.0 × baseline if the baseline was abnormal. ALP elevation was defined as follows: Grade 1 was >ULN-2.5 × ULN if the baseline was normal; 2.0–2.5 × baseline if the baseline was abnormal. Grade 2 was >2.5–5.0 × ULN if the baseline was normal; >2.5–5.0 × baseline if the baseline was abnormal. Grade 3 was >5.0–20.0 × ULN if the baseline was normal; >5.0–20.0 × baseline if the baseline was abnormal. γ-GT elevation was defined as follows: Grade 1 was >ULN-2.5 × ULN if the baseline was normal; 2.0–2.5 × baseline if the baseline was abnormal. Grade 2 was >2.5–5.0 × ULN if the baseline was normal; >2.5–5.0 × baseline if the baseline was abnormal. Grade 3 was >5.0–20.0 × ULN if the baseline was normal; >5.0–20.0 × baseline if the baseline was abnormal. As a treatment for GP-induced liver injury, a dose of 600 mg/day of UDCA was additionally administered without discontinuation of GP therapy at the discretion of the attending physician.

### 2.4. Assessment of Efficacy of Add-On UDCA for GP-Induced Liver Injury

To evaluate the efficacy of add-on UDCA for GP-induced liver injury, changes in TB after onset were compared between add-on UDCA and no UDCA groups in patients with grade ≥ 2 TB elevation, excluding patients with therapy discontinuation, dose reduction, or onset at the end of treatment.

### 2.5. Statistical Analysis

Values are expressed as median and range. Values of *p* < 0.05 were considered significant for analysis. Factors contributing to GP-induced liver injury were analyzed using logistic regression analysis. Factors showing a result of *p* < 0.05 in univariable analyses were entered into the multivariable analysis to identify independent factors. Significant TB changes were tested using the Wilcoxon signed-rank test. In comparisons of the amount of TB change between add-on UDCA and no UDCA groups, the Mann–Whitney U-test was used for analysis. All analyses were performed using SPSS version 24.0 software (SPSS, Chicago, IL, USA).

## 3. Results

### 3.1. Patient Characteristics

Characteristics of patients are summarized in Table 1. The median age was 70 years (range 14–94 years). The median body weight was 58 kg (range 29.6–140.5 kg). Of the 236 patients, 85 patients (36%) were ≥75 years of age, 78 patients (33%) showed cirrhosis, and the ALBI grade was ≥2 in 82 patients (35%).

### 3.2. SVR and GP-Induced Liver Injury

The SVR rate in this study was 99.6% (235/236) in the per-protocol analysis. Incidence, grade, time to onset, and management of GP-induced liver injury are summarized in Table 2. The frequency of GP-induced liver injury was 61.9% (146/236). Among the 146 patients, 10 patients (4.2%) were grade ≥ 3. Incidences of TB, AST, ALT, ALP, and γ-GT elevation were 50% (118/236), 2.9% (7/236), 5.9% (14/236), 25.4% (60/236), and 1.7% (4/236), respectively, and serious elevation to grade ≥ 3 was identified in 3.8% (9/236), 0%, 0%, 0%, and 0.4% (1/209), respectively. Management of GP-induced liver injury comprised therapy discontinuation for 1 patient, a reduction in the dose of GP for 1 patient, and the addition of UDCA to GP for 21 patients due to hyperbilirubinemia. The two patients with dose reduction or therapy discontinuation achieved SVR.

### 3.3. Risk Factors Contributing to GP-Induced Liver Injury

Uni- and multivariate analyses of pre-treatment factors contributing to GP-induced liver injury and grade ≥ 2 GP-induced liver injury are summarized in Table 3 and Table 4. Univariate analyses revealed age, height, weight, body mass index (BMI), white blood cell and platelet counts, ALT, γ-GT, PT, albumin, TB, and ALBI grade were significant factors contributing to GP-induced liver injury. Among these significant factors, age and TB remained independent factors after multivariate analysis. On univariate analyses for factors contributing to grade ≥ 2 GP-induced liver injury, the significant factors were age, weight, BMI, cirrhosis, platelet count, PT, albumin, TB, and ALBI grade. On multivariate analysis, TB was the only independent factor.

### 3.4. Efficacy of Add-On UDCA for GP-Induced Liver Injury

Among 43 patients with hyperbilirubinemia grade ≥ 2, 1 patient with treatment discontinuation, 1 patient with dose reduction, and 2 patients who developed hyperbilirubinemia grade ≥ 2 at the end of GP therapy were excluded from the analysis. Changes to TB in patients with hyperbilirubinemia grade ≥ 2 treated with and without add-on UDCA are shown in Table 5 and Figure 1. Although no significant change in TB was seen in the UDCA group, TB in the add-on UDCA group was significantly decreased during GP treatment. A comparison of the amount of TB change between groups with and without add-on UDCA is shown in Table 6 and Figure 2. The amount of TB depletion was significantly larger in the add-on UDCA group than in the no UDCA group.

## 4. Discussion

This retrospective cohort study was conducted to establish how best to deal with GP-induced liver injury, which occasionally becomes serious and can lead to therapy discontinuation or liver failure. The present study indicated the frequency, time to onset, type of liver injury, and risk factors for GP-induced liver injury. In addition, the efficacy of add-on UDCA for GP-induced liver injury was also shown. The key characteristics of patients in this study were older age and smaller BMI than those in clinical trials. The median age in this study was 70 years old, and the median BMI was 22.8 kg/m^2^. On the other hand, the median age for participants in clinical trials was around 50–60 years [16,17,18,19], and the BMI was around 25 kg/m^2^ [20]. These differences in patient characteristics must therefore be considered when interpreting the present results.

Some meta-analyses, including real-world cohorts, have already reported that the efficacy and safety of GP therapy were similar to those in clinical trials [6,20,21]. The SVR rate was close to 100%, and the adverse event-related discontinuation rate was less than 1%. The efficacy and safety of GP therapy have thus been proven with high-level evidence. This post-marketing observational cohort study was also able to show extremely high efficacy and safety rates, even in the elderly.

As many previous reports of GP therapy did not show minimal hepatic abnormalities like grade 1, the actual incidence of GP-induced therapy is unclear. The frequency of GP-induced liver injury in the present study was higher than expected. In Japanese clinical trials of GP therapy, hyperbilirubinemia was the only grade 3 hepatic abnormality encountered and was found in 2% (1/56) of cirrhotic patients. The incidence of TB elevation grade ≥ 2 in patients with and without cirrhosis was 2.7% (6/219) and 10.7% (6/56), respectively, and that of the cirrhotic patient group was higher than that of the non-cirrhotic patients’ group [1,2]. In a pooled analysis of five phase II/III trials of GP therapy for non-Japanese Asian patients, hepatic abnormalities ≥ grade 3 were rare (0.8%) [19]. However, a systematic review and meta-analysis revealed that GP-induced liver injury occurred frequently, and hyperbilirubinemia occurred the most frequently, particularly in patients with cirrhosis [22]. Furthermore, the meta-analysis also indicated that the most frequent grade 3 treatment-related abnormality among laboratory parameters was TB elevation. Comparing patients with and without cirrhosis, the incidence risk ratio of grade 3 hyperbilirubinemia was high, at 2.724 [22]. In the present study, the most common liver abnormality was again TB elevation and necessitated therapy discontinuation or dose reduction in two patients. Careful attention to bilirubin elevation is therefore warranted during treatment.

Regarding the time to onset for GP-induced liver injury, the most frequent onset time was after 2 weeks, with more than half of GP-induced liver injuries developing within 4 weeks after the start of therapy. Laboratory testing should therefore be performed every 2 weeks during treatment for safety.

The present study analyzed risk factors for GP-induced liver injury. Univariate analysis revealed age, height, weight, BMI, white blood cell and platelet counts, ALT, γ-GT, PT, albumin, TB, and ALBI grade as significant. Although these factors were confounding, elderly, small-sized, cirrhotic patients might logically be considered at high risk of GP-induced liver injury. Among the risk factors identified as significant from univariate analyses, age and TB remained as independent factors after multivariate analysis. Some Japanese real-world cohorts with median ages around 65 years old revealed that the incidence of GP-induced hyperbilirubinemia was within the range of 0.6–2% [23,24,25,26,27,28]. However, Tamori et al. indicated in their real-world cohort that the median age of patients with hyperbilirubinemia was high, at 70 years old [23]. The frequency of TB elevation grade ≥ 3 in the present study was 3.8%, higher than those in previous Japanese real-world cohorts. This is probably attributable to the higher median age in the present study compared to previous Japanese real-world cohorts. Therefore, other DAA regimens without an NS3 protease inhibitor, such as sofosbuvir/velpatasvir, should thus be selected for elderly patients with compensated cirrhosis.

Glecaprevir is mainly metabolized by liver cytochrome P450 (CYP)3A [4,5]. Glecaprevir plasma exposure was increased by 33% in patients with compensated cirrhosis relative to normal subjects [29]. CYP content in the human liver decreases with age, and drug metabolism is reduced by around 30% after 70 years old [30]. In addition, as the dose of GP therapy is not adjusted by body weight, drug plasma concentrations may tend to increase among low-body-weight patients compared to standard-weight patients. Accordingly, glecaprevir plasma concentrations can be expected to be increased in elderly cirrhotic patients compared to non-elderly cirrhotic patients. A reduced dose (e.g., around 30% reduction) of GP may thus be preferable for elderly cirrhotic patients for safety reasons.

In the present study, as TB and ALP elevations were frequently seen in GP-induced liver injury, the main GP-induced liver injuries can be considered to be cholestatic in nature. UDCA is classically used for the treatment of drug-induced cholestatic liver injury [11,12]. In the present study, elevated TB due to GP was significantly decreased in the add-on UDCA group but not when no UDCA was added to GP. This result reflects the efficacy of UDCA for GP-induced hyperbilirubinemia. Glecaprevir is a weak CYP3A inhibitor [5]. On the other hand, UDCA reportedly prevents drug-induced reductions in hepatic CYP isozymes in rats [31]. UDCA may therefore prevent or improve CYP3A enzyme reductions.

Some limitations of the present study should be considered. First, this was a single-center, retrospective cohort study with some biases. Furthermore, the numbers of patients of the add-on UDCA and the no UDCA groups were small. To validate the efficacy of add-on UDCA for GP-induced liver injury, future randomized studies are needed. Second, as no serious elevations in transaminases were seen in the present study, the efficacy of UDCA for GP-induced hepatocellular liver injury remains unknown. Third, from our results, whether UDCA can prevent the onset of GP-induced liver injury is still unknown.

## 5. Conclusions

The most frequent GP-induced liver injury was hyperbilirubinemia, not aminotransferase elevation. Independent risk factors for GP-liver injury were age and TB. Careful attention to GP-induced liver injury is warranted for elderly patients with cirrhosis. As add-on UDCA could suppress the aggravation of GP-induced liver injury, 600 mg/day of UDCA may be added to GP without discontinuing therapy or reducing doses when hyperbilirubinemia of grade ≥ 2 develops. To validate the efficacy of add-on UDCA for GP-induced liver injury, further study is needed.

## Figures and Tables

**Figure 1 viruses-15-00489-f001:**
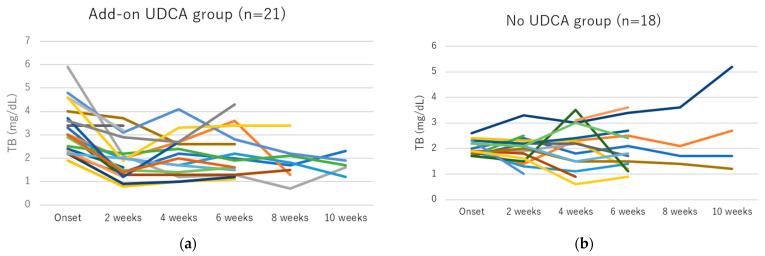
Changes in total bilirubin (TB) for individual patients with hyperbilirubinemia grade ≥ 2. (**a**) TB decreased significantly in the add-on ursodeoxycholic acid (UDCA) group (*p* < 0.05). (**b**) No significant change in TB was seen in the no UDCA group. Change of each patients’ TB value was expressed using different color lines.

**Figure 2 viruses-15-00489-f002:**
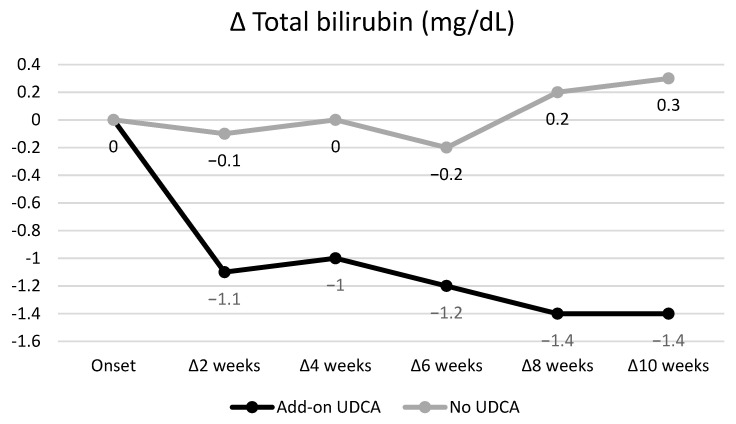
Comparison of the amount of total bilirubin (TB) changes between groups with and without add-on ursodeoxycholic acid (UDCA). The amount of TB depletion was significantly larger in the add-on UDCA group than in the no UDCA group.

**Table 1 viruses-15-00489-t001:** Baseline characteristics of patients.

	N = 236
Age, years (range)	70 (14–94)
≥75 years	85 (36%)
Sex, male/female	132/104
Height, cm (range)	161.0 (133.2–180)
Weight, kg (range)	58 (29.6–140.5)
Body mass index, kg/m^2^ (range)	22.8 (15.6–58.5)
Cirrhosis	78 (33%)
Genotype (1/2/3/4/5/6/unknown)	132/101/1/0/0/1/1
HCV-RNA (logIU/mL)	6.3 (2.1–7.6)
History of HCC treatment	28 (12%)
History of IFN-based therapy	24 (10%)
History of DAA treatment	13 (6%)
White blood cells (/mm^3^)	5180 (1200–11,700)
Hemoglobin (g/dL)	13.7 (7.8–17.3)
Platelets (×10^4^/mm^3^)	17.9 (5.1–96.7)
AST (U/L)	39 (13–432)
ALT (U/L)	36 (8–558)
ALP (U/L)	87 (35–303)
γ-GT (U/L)	37 (7–748)
PT (%)	97 (12–141)
Alb (g/dL)	4.1 (2.3–5.2)
TB (mg/dL)	0.8 (0.3–2.6)
ALBI grade (1/2/3)	154/80/2

Values are expressed as median (range) or number. HCV, hepatitis C virus; DAA, direct-acting antiviral agent; IFN, interferon; HCC, hepatocellular carcinoma; AST, aspartate aminotransferase; ALT, alanine aminotransferase; ALP, alkaline phosphatase; γ-GT, γ-glutamyl transferase; PT, prothrombin time; Alb, albumin; TB, total bilirubin; ALBI, albumin-bilirubin grade.

**Table 2 viruses-15-00489-t002:** Incidence, grade, time to onset, and management of glecaprevir/pibrentasvir-induced liver injury.

	N = 236
Occurrence of liver dysfunction	146 (62%)
Liver injury grade	
AST elevation (Grade 1/2/≥3)	7 (6/1/0)
ALT elevation (Grade 1/2/≥3)	14 (13/1/0)
ALP elevation (Grade 1/2/≥3)	60 (60/0/0)
γ-GT elevation (Grade 1/2/≥3)	4 (2/1/1)
TB elevation (Grade 1/2/≥3)	118 (75/34/9)
Time to liver injury	
AST elevation after 2/4/6/8/10/12 weeks	2/2/2/0/0/1
ALT elevation after 2/4/6/8/10/12 weeks	8/1/2/1/1/1
ALP elevation after 2/4/6/8/10/12 weeks	31/14/6/7/1/1
γ-GT elevation after 2/4/6/8/10/12 weeks	2/1/0/0/1/0
TB elevation after 2/4/6/8/10/12 weeks	94/12/7/5/0/0
Management of liver injury	
Treatment discontinuation due to hyperbilirubinemia	1
Dose reduction due to hyperbilirubinemia	1
Add-on UDCA for hyperbilirubinemia	21

AST, aspartate aminotransferase; ALT, alanine aminotransferase; ALP, alkaline phosphatase; γ-GT, γ-glutamyl transferase; TB, total bilirubin; UDCA, ursodeoxycholic acid.

**Table 3 viruses-15-00489-t003:** Uni- and multivariate analyses of pre-treatment factors contributing to glecaprevir/pibrentasvir-induced liver injury.

	Univariate			Multivariate		
Factors	*p*	OR	95% CI	*p*	OR	95% CI
Age (per 1-year increase)	<0.001	1.056	1.033–1.079	0.005	1.044	1.013–1.076
Sex (female)	0.209	1.408	0.826–2.402			
Height (per 1-cm increase)	0.005	0.960	0.932–0.987	0.614	1.028	0.923–1.145
Weight (per 1-kg increase)	0.002	0.971	0.953–0.990	0.351	0.939	0.822–1.072
BMI (per 1-kg/m^2^ increase)	0.040	0.939	0.885–0.997	0.341	1.179	0.840–1.655
Cirrhosis	0.103	1.614	0.908–2.870			
History of HCC treatment	0.133	1.992	0.810–4.896			
History of IFN-based therapy	0.343	1.563	0.621–3.930			
History of DAA treatment	0.085	0.363	0.115–1.148			
HCV-RNA (per 1-logIU/mL increase)	0.528	1.090	0.834–1.424			
White blood cells (per 1/mm^3^ increase)	0.043	1.000	1.000–1.000	0.829	1.000	1.000–1.000
Hemoglobin (per 1-g/dL increase)	0.139	0.893	0.768–1.037			
Platelets (per 1 × 10^4^/mm^3^ increase)	0.008	0.946	0.908–0.986	0.623	0.989	0.946–1.034
AST (per 1-IU/L increase)	0.175	0.996	0.990–1.002			
ALT (per 1-IU/L increase)	0.039	0.995	0.991–1.000	0.659	0.999	0.992–1.005
ALP (per 1-IU/L increase)	0.914	1.000	0.993–1.006			
γ-GT (per 1-IU/L increase)	0.016	0.996	0.993–0.999	0.093	0.997	0.993–1.001
PT (per 1% increase)	0.022	0.979	0.962–0.997	0.305	0.989	0.969–1.010
Alb (per 1-g/dL increase)	0.013	0.467	0.256–0.852	0.676	1.302	0.377–4.492
TB (per 1-mg/dL increase)	<0.001	10.668	3.640–31.267	<0.001	14.700	3.941–54.822
ALBI grade (≥2)	0.001	2.836	1.552–5.184	0.482	1.501	0.484–4.653

OR, odds ratio; CI, confidence interval; BMI, body mass index; HCC, hepatocellular carcinoma; IFN, interferon; DAA, direct-acting antiviral agent; HCV, hepatitis C virus; AST, aspartate aminotransferase; ALT, alanine aminotransferase; ALP, alkaline phosphatase; γ-GT, γ-glutamyl transferase; PT, prothrombin time; Alb, albumin; TB, total bilirubin; ALBI, albumin-bilirubin grade.

**Table 4 viruses-15-00489-t004:** Uni- and multivariate analyses of pre-treatment factors contributing to ≥grade 2 glecaprevir/pibrentasvir-induced liver injury.

	Univariate			Multivariate		
Factors	*p*	OR	95% CI	*p*	OR	95% CI
Age (per 1-year increase)	0.009	1.038	1.010–1.068	0.614	1.010	0.974–1.047
Sex (female)	0.747	0.896	0.459–1.749			
Height (per 1-cm increase)	0.066	0.969	0.937–1.002			
Weight (per 1-kg increase)	0.024	0.970	0.944–0.996	0.575	0.984	0.931–1.040
BMI (per 1-kg/m^2^ increase)	0.045	0.911	0.831–0.998	0.683	0.963	0.803–1.154
Cirrhosis	0.006	2.563	1.306–5.028	0.195	1.771	0.746–4.205
History of HCC treatment	0.136	1.977	0.806–4.847			
History of IFN-based therapy	0.448	0.614	0.175–2.161			
History of DAA treatment	0.332	0.359	0.045–2.839			
HCV-RNA (per 1-logIU/mL increase)	0.106	0.773	0.566–1.056			
White blood cells (per 1/mm^3^ increase /mm^3^)	0.064	1.000	1.000–1.000			
Hemoglobin (per 1-g/dL increase)	0.080	0.852	0.712–1.019			
Platelets (per 1 × 10^4^/mm^3^ increase)	0.026	0.938	0.886–0.992	0.554	0.980	0.920–1.045
AST (per 1-IU/L increase)	0.789	0.999	0.991–1.007			
ALT (per 1-IU/L increase)	0.312	1.004	0.996–1.011			
ALP (per 1-IU/L increase)	0.914	1.000	0.993–1.006			
γ-GT (per 1-IU/L increase)	0.335	0.998	0.993–1.002			
PT (per 1% increase)	0.019	0.976	0.957–0.996	0.157	0.985	0.963–1.006
Alb (per 1-g/dL increase)	0.023	0.451	0.227–0.896	0.958	1.033	0.313–3.411
TB (per 1-mg/dL increase)	0.001	5.024	1.932–13.069	0.004	5.132	1.679–15.683
ALBI grade (≥2)	0.001	2.834	1.494–5.377	0.651	1.320	0.396–4.392

OR, odds ratio; CI, confidence interval; BMI, body mass index; HCC, hepatocellular carcinoma; IFN, interferon; DAA, direct-acting antiviral agent; HCV, hepatitis C virus; AST, aspartate aminotransferase; ALT, alanine aminotransferase; ALP, alkaline phosphatase; γ-GT, γ-glutamyl transferase; PT, prothrombin time; Alb, albumin; TB, total bilirubin; ALBI, albumin-bilirubin grade.

**Table 5 viruses-15-00489-t005:** Changes in total bilirubin in patients with hyperbilirubinemia ≥ grade 2 treated with and without add-on ursodeoxycholic acid.

	Add-On UDCA Group (*n* = 21)		No UDCA Group (*n* = 18)	
	Total Bilirubin (mg/dL)	*p*	Total Bilirubin (mg/dL)	*p*
At onset	3.0 (1.9–5.9)		2.0 (1.6–3.5)	
After 2 weeks	1.9 (0.8–3.7)	<0.001	2.0 (1.0–3.3)	0.690
After 4 weeks	2.1 (1.0–4.1)	<0.001	2.2 (0.6–3.5)	0.950
After 6 weeks	1.6 (1.1–4.3)	<0.001	1.8 (0.8–3.6)	0.363
After 8 weeks	1.8 (0.7–3.4)	0.012	1.9 (1.4–3.6)	0.500
After 10 weeks	1.7 (1.0–3.5)	0.018	1.9 (1.2–5.2)	0.465

Values are expressed as median (range) or number. UDCA, ursodeoxycholic acid.

**Table 6 viruses-15-00489-t006:** Comparison of total bilirubin changes between with and without add-on ursodeoxycholic acid groups.

Δ Total Bilirubin	Add-On UDCA Group (*n* = 21)	No UDCA Group (*n* = 18)	*p*
After 2 weeks	−1.1 (−3.8–0.0)	−0.1 (−1.3–0.8)	<0.001
After 4 weeks	−1.0 (−4.7–−1.0)	0.0 (−1.3–1.8)	<0.001
After 6 weeks	−1.2 (−4.6–0.7)	−0.2 (−1.5–0.8)	0.004
After 8 weeks	−1.4 (−5.2–−0.4)	0.2 (−0.4–1.8)	0.002
After 10 weeks	−1.4 (−4.3–−0.5)	0.3 (−0.6–2.6)	0.012

Values are expressed as median (range) or number. UDCA, ursodeoxycholic acid.

## Data Availability

The database for the current study is available from the corresponding author upon reasonable request.

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
