# Peer review of "Risk Factors of Glecaprevir/Pibrentasvir-Induced Liver Injury and Efficacy of Ursodeoxycholic Acid"

_viruses, 2023, doi:10.3390/v15020489_

Round 1

Reviewer 1 Report

This is a retrospective cohort study that has been made in a real-world setting. A total of 236 patients with hepatitis C have been treated with the antiviral combination (glecaprevir/pibrentasvir); the frequency of sustained virological response (SVR rate) was 99.6% (235/236) according to a per-protocol analysis. The authors found that 146 of 236 patients had GP-related liver damage. Management of GP-associated liver toxicity included the addition of ursodeoxycholic acid (UDCA) in 21 patients with high values of hyperbilirubinemia. Multivariate analysis was carried out; age and total bilirubin values had an independent and significant relationship with liver damage by GP. Therapy with UDCA lowered consistently bilirubin levels whereas no changes in bilirubin values resulted in the no UDCA patient group.

Some points of the paper are unclear:

1) The authors enrolled 236 patients with hepatitis C. Please let us know how many had chronic or acute HCV;

2) The add-on UDCA and the no UDCA groups were small (n=21 and n=18 patients, respectively). This is an important shortcoming of the study;

3) The authors have included no reference with concern to the criteria for the definition of GP-induced liver toxicity. In addition, they have not clearly mentioned (at least shortly) the criteria for the definition of AST (or ALT, FA, GGT) elevation (grade 1, grade 2, or grade 3);

4) The issue addressed by the authors has hot nature as GP antiviral therapy is a common option for the treatment of HCV and HCV remains common all over the world;

5) The study is not original as other papers have been published on the topic; some of these papers have been appropriately included in the Section References of the manuscript

6) The criteria for the definition of 'GP-induced liver toxicity' remain unclear, especially to the busy readers

Author Response

Responses to Comments from Reviewer 1

This is a retrospective cohort study that has been made in a real-world setting. A total of 236 patients with hepatitis C have been treated with the antiviral combination (glecaprevir/pibrentasvir); the frequency of sustained virological response (SVR rate) was 99.6% (235/236) according to a per-protocol analysis. The authors found that 146 of 236 patients had GP-related liver damage. Management of GP-associated liver toxicity included the addition of ursodeoxycholic acid (UDCA) in 21 patients with high values of hyperbilirubinemia. Multivariate analysis was carried out; age and total bilirubin values had an independent and significant relationship with liver damage by GP. Therapy with UDCA lowered consistently bilirubin levels whereas no changes in bilirubin values resulted in the no UDCA patient group.

Some points of the paper are unclear:

1) The authors enrolled 236 patients with hepatitis C. Please let us know how many had chronic or acute HCV;

→No patient with acute hepatitis C was included in this study.(page 2 line 62-63)

2) The add-on UDCA and the no UDCA groups were small (n=21 and n=18 patients, respectively). This is an important shortcoming of the study;

→Following reviewer’s advice, this limitation was noted in discussion section.(page 10, line 274-275)

3) The authors have included no reference with concern to the criteria for the definition of GP-induced liver toxicity. In addition, they have not clearly mentioned (at least shortly) the criteria for the definition of AST (or ALT, FA, GGT) elevation (grade 1, grade 2, or grade 3);

→Given the  reviewer’s comment, we cited another reference for definition of GP-induced liver injury. In addition, we also added the grading criteria of CTCAE version 5. (page 2, line 80-81, and line 85-101)

4) The issue addressed by the authors has hot nature as GP antiviral therapy is a common option for the treatment of HCV and HCV remains common all over the world;

→Thank you for your comment.

Reviewer 2 Report

I would just like the authors to add some comments regarding a few points:

1. The point on the correlation between tuberculosis and DAA should be explored, which mechanism would favor the onset of tuberculosis in those who take these drugs.

2. The conclusions should be further investigated.

I do not have other comments to add for the authors. I appreciate this work and I think will take an added value to the Journal. 

Author Response

Responses to Comments from Reviewer 2

I would just like the authors to add some comments regarding a few points:

  1. The point on the correlation between tuberculosis and DAA should be explored, which mechanism would favor the onset of tuberculosis in those who take these drugs.

→Thank you for your comment. There was no patient who had the onset of tuberculosis after GP therapy in this study.

  1. The conclusions should be further investigated.

→Given the reviewer’s advice, we added the necessity of further investigation in conclusions section.(page 10, line 287-288)

I do not have other comments to add for the authors. I appreciate this work and I think will take an added value to the Journal.
